# Development and Performance Evaluation of Integrated Hybrid Power Module for Three-Phase Servo Motor Applications

**DOI:** 10.3390/mi14071356

**Published:** 2023-06-30

**Authors:** Hsien-Chie Cheng, Yan-Cheng Liu, Hsin-Han Lin, Shian-Chiau Chiou, Chih-Ming Tzeng, Tao-Chih Chang

**Affiliations:** 1Department of Aerospace and Systems Engineering, Feng Chia University, Taichung 407, Taiwan; 2Ph.D. Program of Mechanical and Aeronautical Engineering, Feng Chia University, Taichung 407, Taiwan; 3WBG Device & Module Design Department, Industrial Technology Research Institute, Hsinchu 300, Taiwan

**Keywords:** integrated hybrid power module, SiC MOSFET inverter, space-vector pulse width modulation, power conversion efficiency, electromagnetic-circuit co-simulation, cancellation circuit, double pulse test, inductance measurement, open-loop inverter operation

## Abstract

This study aims to develop a 30 kHz/12 kW silicon carbide (SiC)/Si integrated hybrid power module (iHPM) for variable frequency drive applications, particularly industrial servo motor control, and, additionally, to theoretically and experimentally assess its dynamic characteristics and efficiency during operation. This iHPM integrates a brake circuit, a three-phase Si rectifier, and a three-phase SiC inverter within a single package to achieve a minimal current path. A space-vector pulse width modulation (SVPWM) scheme is used to control the inverter power switches. In order to reduce parasitic inductance and power loss, an inductance cancellation design is implemented in the Si rectifier and SiC inverter. The switching transients and their parasitic effects during a three-phase operation are assessed through an electromagnetic-circuit co-simulation model, by which the power loss and efficiency of the iHPM are estimated. The modeled parasitic inductance of the inverter is validated through inductance measurement, and the effectiveness of the simulated results in terms of switching transients and efficiency is verified using the experimental results of the double pulse test and open-loop inverter operation, respectively. In addition, the power loss and efficiency of the SiC MOSFET inverter are experimentally compared against those of a commercial Si IGBT inverter.

## 1. Introduction

As market demand evolves, the future development of power modules (PMs) must not only meet system specifications, but also achieve high efficiency, high switching frequency, and high reliability under high operating temperature [1]. Many PM-related applications, on the one hand, have space constraints (e.g., aircraft, EVs, unmanned vehicles, robotic arms), and on the other hand, require high power and high frequency (e.g., DC–DC converters, AC–DC–AC variable frequency drives (VFD), uninterruptible power supplies). This would likely lead to a high-power density and excessive temperatures, which may successively cause higher power loss, eventually going into thermal runaway and reliability issues [2]. It is evident that high power density PMs face many challenges, such as the specifications of the system requirements for high power applications, the selection of power semiconductor devices (PSDs), low inductance circuit designs, system layout designs, thermal management, and even reliability. The performance and specification requirements for PMs and PSDs differ among various applications [3]. With the advancement of material and process technologies, the use of wide bandgap (WBG) SiC and gallium nitride (GaN) as semiconductor materials for high-power and high-frequency transistors has become common practice because of their many benefits over conventional Si materials, such as lower power loss, higher breakdown voltages, higher operating electric fields, higher operating temperatures, and higher switching frequencies [4,5,6,7,8,9]. Under high voltage and high temperature operating conditions, it was found [8] that the performance of SiC PSDs is far superior to that of Si PSD. In addition, WBG materials can improve efficiency, reduce size and weight, and lower system costs [10].

SiC devices possess high current rates and therefore faster switching capabilities, compared with Si ones, making them more sensitive to module encapsulation, parasitic parameters, and gate drive parameters. For instance, large voltage spikes occur during switching transients due to short pulses with a high current rate when the switching frequency is high. In addition, parasitic inductance will resonate with the parasitic capacitance, resulting in increased electromagnetic interference (EMI) and disturbance ringing [11]. Moreover, parasitic inductance may bring about significant overshoots and ringing in the current and voltage waveforms, which would, in turn, result in increased EMI. In this respect, PSDs demand a higher voltage rating than what is needed. Consequently, the parasitic inductances in both the power and gate circuits must be minimized to suppress the induced overvoltage and overcurrent. To alleviate these issues, an RC snubber circuit [12,13], the increased gate resistance (R_g_) of the gate control circuit [14], or even reduced parasitic inductance via electromagnetic suppression/cancellation [15] can be an effective means. However, there is a tradeoff in the former two approaches, as both can lead to additional power loss and reduced efficiency, which would counter the purpose of using the expensive SiC PSDs. By contrast, electromagnetic cancellation via the layout design of PMs could be a more effective way to reduce parasitic inductance, where three-dimensional [15,16,17,18,19] and planar [12] architectures have been proposed as effective solutions for reducing parasitic inductance and even enhancing thermal performance. For instance, it was revealed [18] that magnetic field cancellation between two terminals, as a result of the P-N terminal overlap arrangement in the unique structure of a 3D-stacked module, can lead to an exceptional decrease in parasitic inductance. Moreover, incorporating the PSDs into the power circuits in such a way that unwanted parasitic inductances can be removed is a feasible alternative for achieving low parasitic inductance. Using this approach, Chen et al. [19] proposed a SiC-based, half-bridge module with a hybrid packaging method based on the incorporation of the printed circuit board (PCB) and the direct bonded copper (DBC) structure. This structure allows the minimization of the parasitic inductances through the optimization of current communication loops, driver locations, and gating connections. Furthermore, Raab et al. [12] developed a high efficiency, fast-switching, and reliable full-SiC PM based on a novel planar packaging technology for More Electrical Aircraft (MEA) applications through the integration of snubber capacitors, gate drivers, and online temperature measurement solutions for health-monitoring. Additionally, two isolated gate driver ICs were employed to assure decreased and symmetric gate loop inductances, further facilitating symmetric current sharing during the switching transients.

IC substrates and cooling systems are the primary heat dissipation passages of PMs. A proper selection of substrate materials and the use of effective cooling systems are critical to the device’s thermal performance and even thermal–mechanical reliability. DBC alumina (Al_2_O_3_ and AlN) ceramic substrates have been widely used in various power electronics applications due to technology maturity, good thermal and mechanical performances, and even low production costs, especially the Al_2_O_3_ DBCs. As a result of a lack of an oxide layer for copper (Cu) bonding, the production cost for the AlN DBCs is markedly higher than that of the Al_2_O_3_ ones because of the greater manufacturing difficulty and lower yield despite having superior thermal performance. Moreover, these insulators are brittle in nature and thus vulnerable to stress cracking [20,21]. It is worth noting that a thicker insulator causes not only higher electrical insulation but also greater thermal resistance, which is not ideal for high power applications. To address these challenges, several different types of substrates were proposed, including insulated metal substrates (IMS), Si_3_N_4_, direct plate copper (DPC), etc. Among them, IMS is an attractive alternative as a result of its low cost in manufacturing, the good mechanical property of the insulation layer, great efficiency in layout usage, and even superior thermal and electrical performances [21,22]. Their thermal performance can be upgraded through the use of a thinner dielectric layer though this may unfortunately lead to a lower electrical resistance [23]. It was reported that a Cu- and Al-based IMS substrate with an insulating layer (IL) thickness of 0.12 mm outperforms an Al_2_O_3_ DBC substrate with an IL thickness of 0.38 mm in thermal performance [22].

The power loss generated by the internal PSDs of PMs during operation can induce a significant temperature rise, ultimately affecting their performance and reliability. An accurate estimation of their power loss and efficiency during operation is a crucial issue in the development of PMs. In general, the switching loss of PMs tends to increase with the switching frequency, and in addition, temperature would play a significant role in the power loss, especially the conduction loss [24], which may further trigger thermal runaway. During the past years, there have been many reports on switching loss evaluation through theoretical analysis and experimental characterization [14,24,25,26]. For example, Xiong et al. [25] introduced a physically based semiconductor device modeling method to examine the switching loss of a power MOSFET device, and in accordance with the new physical insights, to evaluate the widely used switching loss estimation approach. Qi et al. [26] found that the switching loss of the Si MOSFET and IGBT devices is positively proportional to temperature, while it is the opposite for the SiC MOSFET. Cheng et al. [14] conducted numerical analysis and experimental measurements to investigate the switching transients and power loss of an SOT-227 packaged Si MOSFET PM and the associated three-phase power MOSFET inverter during load cycles. To effectively evaluate the PM electrical behavior and even power loss, Dutta [27] applied an electromagnetic-circuit co-simulation (ECC) approach to analyze the switching response of a three-dimensional stacked wire bondless IGBT power module under various system configurations. Lin et al. [28] investigated the conducted EMI issue on an AC–DC switching power adaptor with a flyback converter topology through ECC.

Although there have been some studies on the parasitic inductance, switching waveforms, and system operation and performance of SiC PMs over the past years, there is still limited development of a comprehensive systematic exploration of Si/SiC integrated hybrid PMs (iHPMs) from PM design to system operation. This study attempts to develop a 30 kHz/12 kW SiC/Si iHPM (Figure 1) for industrial servo motor drives. This iHPM integrates three functions—a brake circuit, a three-phase Si rectifier, and a three-phase SiC inverter within a single package to achieve a minimized current path. An insulated metal substrate (IMS), rather than a DBC substrate, is used to provide structural support to these power devices and components, and to facilitate electrical isolation and thermal transfer from the device to a base plate, a heatsink, and even a cooling system. A special focus of this PM development is placed on the inductance cancellation design in the rectifier and inverter circuit in order to accomplish a reduced parasitic inductance. To control the output voltage of the three-phase inverter, a space-vector pulse width modulation (SVPWM) technique is implemented. The switching characteristics, dynamic behavior, and performance of the iHPM during the three-phase switching operation are characterized experimentally and theoretically. Moreover, in order to demonstrate the performance advantage of the SiC MOSFET inverter system in terms of system power loss and efficiency, the Si IGBT inverter system inside a commercial integrated PM (iPM) is experimentally analyzed.

## 2. Development of Sic/Si Integrated Hybrid Power Module

Power conversion system designs, such as inverters and converters, for small AC motors in industrial applications face increasing needs and also challenges in meeting stringent specification requirements for reliability, size, cost, and energy efficiency. Traditionally, various power conversion designs are fulfilled using discrete power semiconductor packages [29]. Although they have several advantages, such as reduced cost, great flexibility in choice of PSDs, low power, and a simple topology, these packaging approaches would require a relatively much larger and more intricate PCB design to satisfy the constraints of spacing and layout of these discrete power packages and active and passive components. This problem can be properly addressed using an integrated packaging solution, allowing for a reduction in stock holding and handling costs and assembly effort, in comparison with the discrete one.

The prototype of the proposed iHPM is illustrated in Figure 1a, and its electrical layout is presented in Figure 1b, which incorporates three major components, namely a brake circuit, a three-phase rectifier, and a three-phase inverter. For minimized switching loss and increased power conversion effectiveness, twelve SiC power MOSFET devices with SiC body diodes are implemented in the inverter. Furthermore, for cost efficiency, the three-phase rectifier is made up of six Si diodes connected in series and in parallel. These power components are mounted onto an electrically isolated IMS. In addition, this power conversion system is connected to an external AC power supply and a servo motor. Specifically, this iHPM utilizes an AC energy source, rectifies it to a stable DC voltage, converts the DC into three-phase AC through an SVPWM control scheme, and then feeds it into an induction motor based on the demands of load torque and speed. In this AC motor drive system, the three-phase voltage source SiC inverter is the core component in the power module for regulating the voltage level and voltage frequency for load speed control. The output power requirement for the iHPM is 12 kW. It is noted that a gate driver is employed to drive the gate voltage of these power switches of the SiC inverter. Considering the size limits, the gate control unit is not packaged in the iHPM. The design considerations for the three-phase SiC inverter are briefly described in the following. Two parallel-connected SiC power MOSFET devices per switch are used to upgrade the current rating. The output, transfer, and body diode characteristics, measured using a curve analyzer, are reported in Figure 2, and the diode characteristics of the rectifier diodes are depicted in Figure 3. For safety, the module design includes a brake circuit between the three-phase Si rectifier and SiC inverter. The brake circuit implements an Si IGBT device to control the brake and an Si diode to discharge the energy using the external resistor. It is noted that the resistor for energy discharge is not included in this power module. The SiC MOSFETs rated at 1200 V and 149 A at 100 °C, produced by Wolfspeed, and the Si diodes rated at 1800 V and 100 A, manufactured by YENYO technology, are implemented. The maximum junction temperature rating of the power chips is 175 °C.

A substrate is a crucial part of and a determining factor in a power module, that highly affects its electrical, thermal, and mechanical performance. In contrast to the double-sided DBC substrate, IMS is a single-sided substrate, suggesting that only one side (i.e., the Cu side) can be installed with electronic components. IMSs have been extensively reported in the literature on power electronics applications [21,22,30,31,32]. According to the literature data, the thermal performance of an IMS is believed to be typically inferior to AlN DBC substrates mainly due to the low thermal conductivity of their dielectric layer. However, in addition to simplifying the manufacturing process and enabling the feasibility of multi-stacked layers, this technology also allows thermal and electrical isolation performance enhancements through the optimization of the material, the number of layers, and the thickness of the conductive and dielectric layers. For example, the thermal performance of IMS can be upgraded by using a thinner high thermal conductivity thermal interface material (TIM). According to the vendor’s datasheet, the TIM with a thickness of 100 μm possesses a breakdown voltage greater than 6 kV, thereby having the potential to achieve good electrical insulation or prevent current leakage between the conductive layers. In this work, for a better compromise between cost and performances [21], an advanced IMS that applies a polymer-based material as an electrical insulator is employed. This IMS, as reported in Figure 4a, is made up of a 3 mm thick metal Cu baseplate covered by a thin epoxy-based layer of 0.01 mm thickness as a dielectric and a 0.3 mm thick Cu trace layer. During the three-phase switching operation, the IMS is further attached using thermal grease to an Al heatsink with a cooling fan for fast heat dissipation. These twelve power MOSFET devices are mounted on the top Cu islands of the IMS using a SAC305 solder, and then connected via 10 Al bond wires to establish a three-phase bridge power module. In order to protect these bond wires from fusing, the bond wire number is determined based on the maximum loading current. It is worthwhile mentioning that there needs to be a connection wire for facilitating an electrical conduction from the gate pin to the gate pad of the power chip, and from the sense pin to the source side.

Package parasitic inductances can be extensively minimized via a cancellation design by virtue of a proper module layout (substrate pattern) and pin assignment designs. Figure 4b displays the Cu trace layout and pin assignment designs. Noticeably, there are AC input pins (R, S, T) and DC output pins for the three-phase Si rectifier, DC input pins (P, N), and AC output pins (U, V, W) for the three-phase inverter, and an output pin between the IGBT and the diode for external wiring to the resistor for the brake circuit. Moreover, the gate control also needs one control pin and one sense pin for all these inverter power switches, and also one control pin for the Si IGBT device of the brake circuit. The pin assignment must follow the control board design, and also needs to comply with isolation protection standards, such as those set by the International Electrotechnical Commission (IEC), to prevent short circuits caused by differences in high potential. In this study, the pin assignment design is shown in Figure 4b, which matches the design of a state-of-the-art commercial Si IGBT integrated PM. As shown in this figure, the DC input pins, i.e., P and N, and also the DC output pins, i.e., DC+ and DC−, are designed adjacent to each other as much as possible to ensure that the positive and negative current paths in the substrate can be interleaved to reduce the parasitic effects, in accordance with the design rule for inductance cancellation. In this power module, the high side (hereinafter referred to as the “HS”) is near the output pin, and the lower side (hereinafter referred to as the “LS”) is near the DC input pin. The same design logic, in terms of pin assignment and Cu circuit layout, is also applied to the three-phase Si rectifier. The gap between the top Cu islands is about 1 mm. Because the top side of the power module is filled with a molding compound, the 1 mm gap width is sufficient to keep them electrically isolated from each other. The dimension of the proposed 1200 V/200 A SiC/Si iHPM is listed in Table 1. Figure 5 demonstrates the schematic diagram of a cross-sectional view of a portion of the iHPM structure near the SiC inverter.

## 3. Experiments

### 3.1. Double Pulse Test

The DPT is currently the most commonly used method for testing transistor switching characteristics. This method is numerically and experimentally applied to explore the switching characteristics of the SiC power MOSFET devices, and the results are compared with each other. For the DPT experiment, the measurement circuit is depicted in Figure 6. The experimental equipment includes a high-voltage power supply, a high-frequency oscilloscope, a high-voltage differential probe, a function generator, a Rogowski coil current probe, a load inductor, a gate driver board, and a power board. In the experiment, a half-bridge circuit structure is required, principally owing to that one switch needs to act as a freewheeling diode (FWD) to provide a current path for the freewheeling current as the switch is turned off, and the other switch acts as the device under test (DUT) for switch control. The gate of the SiC MOSFET device serving as the FWD is short-circuited to the source, while the gate and source terminals of the DUT are connected to the control board. The gate driver circuit can provide a gate-source voltage (V_gs_) in the range of −4.0~15 V for driving the SiC MOSFET device. Finally, a current probe is connected in series between the DC capacitor and the drain of the DUT, and a high-voltage differential probe is connected in parallel with the DUT, for measuring the source current (I_d_) and the source-drain voltage (V_ds_) of the DUT, respectively. The control signal can be divided into two square waves, where the duration of the first pulse is mainly determined by the rated current and the voltage values, as well as the load inductance. Based on the inductor voltage and the current relationship, the load current is found to be 200 A with a load inductance of 39 nH, the DC bus voltage 600 V, and the pulse width is 75 µs. As soon as the I_d_ reaches the rated current value, the SiC MOSFET device turn-off waveform can be measured by detecting the gate-to-source voltage signal. During the turn-off period, a current path is formed from the load inductor to the body diode, allowing the load current to remain almost constant, ensuring it reaches the rated current as the DUT is switched on during the second square wave. The measured turn-off and turn-on waveforms disclose the switching time and switching characteristics of the SiC MOSFET device. To facilitate the comparison in switching characteristics among different PSDs or PMs, the international standard IEC 60747-8 [33], that define the switching characteristics, can be adopted.

### 3.2. Inductance Measurements

One of the advantage features of this module design is its low parasite inductance. In addition to numerical modeling, the parasitic inductance measurements of two current paths (i.e., from the terminal P to the terminal O and from the terminal O to the terminal N) of the iHPM are conducted for model validation. The measurement follows the IEC 60747-15 guidelines [34]. The measurement circuit is presented in Figure 7a, where *L_m_* is the parasitic inductance of the module. It is noteworthy that the measurement method and the experimental instruments and setup are similar to those of the DPT. The major difference is that the DUT is not limited to a single switch but can also be a half-bridge circuit. As can be seen in Figure 7a, unlike the DPT shown in Figure 6, the DUT is also an FWD; therefore, the gate terminal of the DUT needs to be short-circuited to the source terminal to ensure that the power switch is in the off state. Moreover, another independent power switch is used as the drive unit on the lower arm for a half-bridge circuit. It is worth noting that the switching characteristics and the rated current and voltage of this independent power switch need to be close to the DUT as much as possible. The double-pulse drive signal is applied to measure the voltage step *V_STEP_* and the current slope (*di*/*dt*) (the rate of current change) during the switch turn-on and the body diode reverse recovery transition. The typical waveform schematic diagram is illustrated in Figure 7b. The *V_STEP_*, as a result of the parasitic inductance, can be characterized during a period of time in which the diode has not yet achieved its blocking capability and the current slope remains constant. The relationship among the parasitic inductance, the current slope, and the voltage plateau can be expressed as,
(1)$$L_{m}=\frac{V_{STEP}}{\left|di/dt\right|}$$

### 3.3. Open-Loop Three-Phase Inverter Operation Experiment

The proposed iHPM is used as a VFD for a 12–18 kW servo motor. To characterize the actual output performance of the three-phase inverter, a three-phase inverter operated in open-loop mode is analyzed. The setup of the three-phase inverter operation is displayed in Figure 8. More details can be seen in references to previous works (see, e.g., [35,36]). The main functionality of the experimental system can be divided into four parts: power module, gate driver, power supply, and output stage. To reduce the length of the current path from the DC power supply and the gate driver board to the SiC iHPM in the experimental system, the power board and the gate driver board are integrated into one printed circuit board (PCB), which is called the gate driver power supply board [37]. The hardware of the gate driver circuit comprises a gate driver power supply board, a commercially available current sensing board, and a microcontroller unit (MCU) board. The DC power supply provides a 400 V DC bus voltage connected to the gate driver power supply board. Moreover, the output of the gate driver power supply board is connected in series with an LC filter and a resistive load. The values of the inductance and the capacitance used in the LC filter are 1.25 mH and 20 µF, respectively. The resistive load is composed of fifteen resistors of 60 Ω configured in a delta topology, as presented in Figure 8b, with each phase connected in parallel with five resistors to form a resistance of 12 Ω. It is worth mentioning that these fifteen resistors are placed in a cooling box to prevent the resistive load from overheating. Additionally, a large heat sink and two high-speed cooling fans are used to prevent the operating temperature of the SiC iHPM from rising. The load is a 12 kW power inverter with a switching frequency of 15 kHz, a modulation index 𝑚 = 1.15, a duty cycle of 82%, and an output frequency of 60 Hz with 120 degrees of phase shift. Notably, a 2.0 µs dead time delay is implemented to prevent the high-side and low-side power switches from cross conduction. To improve system stability during the operation, an external resistance of 13 Ω is applied on the gate driver board.

## 4. Methodology

### 4.1. Power Loss Estimation

The SiC inverter and Si rectifier inside the iHPM generate power loss during the operation. The power loss of the inverter mainly includes switching loss, conduction loss, and current leakage loss of the power MOSFETs, and the conduction and reverse recovery losses of the body diodes. The current leakage loss is typically much lower than the other two [38], and thus can be ignored. The switching loss (*P_sw_*) of a power MOSFET occurs in the region where the voltage and current overlap during the switching transients. It can be assessed based on the turn-on energy loss (*E_on_*), turn-off energy loss (*E_off_*), and switching frequency (*f_s_*) as:(2)$$P_{sw}=(E_{ON}+E_{OFF})\times f_{s}$$

The definition of the *E_on_* and *E_off_* can be found in Refs. [13,14,33]. Furthermore, the conduction loss *P*_*Con*_ of a power MOSFET is the product of conduction current and voltage when the power MOSFET is in the on state,
(3)$$P_{Con}=I_{ON}\cdot V_{ON}$$
where *I_on_* is the current during conduction and *V_on_* is the voltage during conduction.

Body diodes during operation can primarily produce conduction loss and reverse recovery loss. Due to their relatively poor performance, as compared with external Schottky diodes, body diodes may generate much more significant conduction and reverse recovery losses more than the Schottky diodes. Typically, when the high-side power switch turns off, the electrical current travels through the body diode of the complementary low-side power switch, thus resulting in the body diode conduction loss [13]. On the other hand, the body diode reverse recovery loss principally originates from the discharge of the charge kept in the parasitic capacitance of the body diode when the high-side power switch turns on. This loss is dependent on the reverse recovery characteristics of the body diode, namely the body diode reverse-recovery charge and the reverse-recovery voltage across the body diode [14].

### 4.2. Space Vector Pulse Width Modulation

Though SPWM schemes have been broadly applied for VFD applications, such as converters and inverters, the maximum amplitude of the sinusoidal waveforms, also termed the maximum linear modulation index, is only 1.0, so that each phase leg of the inverter can only provide a leg voltage of the amplitude V_dc_/2. If the modulation index is larger than 1, the PWM inverter will operate in the overmodulation region, and the output voltage will still be clamped to ±V_dc_/2, resulting in a substantial decrease in the output AC voltage quality and DC bus utilization. Today, despite computational complexity and intensity, SVPWM is a more preferred and ideal PWM scheme due to benefits, such as its capability in potentially having a greater output voltage without distortion of the waveforms of the line-to-line output voltage, its low harmonic content, and its fixed switching frequency [39,40]. In order to improve its computational complexity and inefficiency, a simpler and more computationally effective SVPWM scheme has been developed through injecting a triangular zero-sequence signal to the sinusoidal waveforms (SPWM) to create non-sinusoidal modulation waveforms. This approach is also known as zero-sequence injection PWM [41]. The non-sinusoidal modulation waveforms would possess a lower peak voltage than SPWM, and more importantly, the modulation index can be further maximized, so that the output voltage can reach the limit of the DC bus voltage. According to the literature report [41], the maximum linear modulation index is 1.15. The calculation principle is depicted as follows, assuming that there is a virtual neutral point *o* in the DC link voltage (as shown in Figure 9a), the voltages of phase *U* and phase *V* relative to point *o* can be represented as:(4)$$V_{Uo}=V_{Un}+V_{no}$$
(5)$$V_{Vo}=V_{Vn}+V_{no}$$
(6)$$V_{UV}=V_{Un}-V_{Vn}$$
where *V_no_* is the phase voltage difference between the neutral point *n* and the virtual neutral point *o*. Equation (6) indicates that *V_no_* can be removed when calculating the line voltage *V_UV_*. The *V_no_* can be used in enhancing the power module voltage usage rate. Following is the voltage range of each phase relative to point *o*:(7)$$\frac{-V_{d}}{2}\leq V_{Uo}\leq \frac{V_{d}}{2}$$

The range of *V_Uo_* voltage can be, therefore, obtained from Equation (4) as follows:(8)$$\frac{-V_{d}}{2}\leq V_{Un}+V_{on}\leq \frac{V_{d}}{2}$$
(9)$$\frac{-V_{d}}{2}-V_{Un}\leq V_{on}\leq \frac{V_{d}}{2}-V_{Un}$$

Considering the voltage range of *V_Un_*, Equation (9) can be rewritten as,
(10)$$\frac{-V_{d}}{2}-V_{\min}\leq V_{on}\leq \frac{V_{d}}{2}-V_{\max}$$
where
(11)$$V_{\max}=Max(V_{Un},V_{Vn},V_{Wn})$$
(12)$$V_{\min}=Min(V_{Un},V_{Vn},V_{Wn})$$

*V_max_* and *V_min_* in Equations (11) and (12), respectively, represent the maximum and minimum voltage values of each phase within one cycle. *V_no_* can be termed zero-sequence injection if it is set as,
(13)$$V_{no}=-\frac{1}{2}\left(V_{\max}+V_{\min}\right)$$

Substituting Equation (13) into Equations (4)–(6), and multiplying the coefficient m yield, the control voltage waveform of zero-sequence injection PWM,
(14)$$v_{si}=m\left(V_{si}-\frac{1}{2}\left(V_{i,\max}+V_{i,min}\right)\right)\;(i=U,V,W),$$
where *V_si_* denotes the control voltage calculated with the original control circuit, and *v_si_* the control voltage after zero-sequence injection. It is noted that the coefficient m is used to compensate for the loss of signals produced by the calculation. The waveforms of phases *U*, *V*, and *W* are illustrated in Figure 9b.

## 5. Numerical Modeling

### 5.1. Electromagnetic Analysis

A 3D quasi-static electromagnetic analysis (EA) is performed using the ANSYS^®^ Q3D extraction tool for extracting the parasitic parameters of the internal current paths of the SiC/Si iHPM. The EA employs finite element methods to establish the parasitic matrix induced using the internal current paths for use in the subsequent solution of the electromagnetic circuit coupling problem. The resulting matrix of the parasitic parameters can be used to generate a module sub-equivalent circuit (MSEC) model for characterizing the parasitic effect on switching transients and power loss. Notably, the MSEC model takes into account both the self- and mutual-effects of the parasitic parameters. In addition, an adaptive meshing scheme is applied in EA for the adaptive adjustment/refinement of the mesh. The parasitic parameter extraction for the specified paths can be conducted by assigning the source and the sink of the current paths. There are three main current paths in the inverter circuit: (1) the positive terminal to the output terminal (P-O), (2) the output terminal to the negative terminal (O-N), and (3) the positive terminal to the negative terminal (P-N). The rectifier circuit resembles the inverter circuit in the circuit structure but with an opposite current direction. The input terminals of the rectifier circuit are the AC terminals of the three phases (R, S, and T), and the output terminals are the positive and negative DC terminals. Si diodes are used in the rectifier circuit. In the circuit structure, the current path of each phase is divided into two distinct path segments: one connects the positive terminal to the AC input terminal, and the other connects the negative terminal to the AC input terminal. The rising time of the time domain signal [14] should be used to specify the maximum frequency, as illustrated in the following equation,
(15)$$F_{MAX}=\frac{1}{t_{on}}.$$

The geometric model for the EA of the SiC/Si iHPM is displayed in Figure 10, which consists of the SiC MOSFET chips, Al wires, Al pads, Cu terminals, an IMS with a Cu baseplate and traces, and die-attach layers. The EA model is composed of 848,419 elements. Figure 10a reveals the current paths in the three phases (U, V, and W) of the inverter and three phases (R, S, and T) of the rectifier, respectively, and Figure 10b presents the circuit of the iHPM with the parasitics. The parasitic inductances of the P-O and O-N current paths in the inverter are validated through the parasitic inductance measurement. In order to facilitate the co-simulation between EA and circuit analyses, the current path of the inverter of each phase can be further divided into four segments by referencing these power switches as the data. Take the U phase as an example: it includes the positive terminal to the drain of the upper arm switch, the output terminal to the source of the upper arm switch, the output terminal to the drain of the lower arm switch, and the source terminal of the lower arm switch to the negative terminal. The remaining two phases of the inverter have the same half-bridge circuit as explained earlier, resulting in a total of twelve current path segments. It is also important to account for the current paths from the gate terminals to the gate of the power switches in the inverter. Likewise, by using the rectifier diodes as the data, four current path segments are established for the current path of the rectifier of each phase due to the positive and negative terminals of the diode. In total, there are thirty current path segments in the iHPM. In addition, the EA model is also applied to assess if the electric current passing through the conductors of the SiC/Si iHPM would exceed its permissible current-carrying capacity.

### 5.2. Electromagnetic-Circuit Co-Simulation (ECC)

To accurately investigate the switching transient behavior of the SiC MOSFET and power loss of the SiC/Si iHPM, an ECC model is established. To comprehensively describe the switching transients of the SiC MOSFETs, the device models of the SiC MOSFET and the Si rectifier diode incorporating the characteristic curves and rated values are established. Subsequently, the characterized parasitic matrix from the EA analysis is employed to form an MSEC model for circuit analysis, and the MSEC is further combined with the developed device models to establish the ECC model. The ECC model is depicted in Figure 11, including the inverter circuit and the Si/SiC iHPM circuit. In addition, as can be seen in the enlarged view of a power switch circuit in Figure 11a, the parasitic capacitances resulting from the SiC MOSFET devices and their nonlinear dependence on the drain-to-source voltage (V_ds_) are considered in the ECC model.

One of the phases of the inverter circuit is used as the DPT circuit, as presented in Figure 11a. Since the positive and negative terminals of the iHPM are shared by these three-phase bridges, there may be issues with the mutual inductance effect on the current paths between each phase. Thus, the inverter ECC model is also used for calculating the switching transient and power losses of the DUT in the DPT circuit. The external gate resistance of 3 Ω and an inductive load of 34 μH are applied without a three-phase resistive load. The DPT analysis is carried out at a 600 V DC bus voltage and a 200 A load current under room temperature. The primary difference between the ECC model of the DPT and the inverter lies in the load connection and control signal, where the latter is shown in Figure 11b. As mentioned previously, the delta-connected resistive load of 12 Ω/phase, connected in series with the three-phase output of the MSEC model, is adopted in the inverter ECC model. The LC filter between the inverter and the resistive load is also implemented. The operation conditions in the present simulations are the same as those described in the experimental system.

Figure 11c illustrates the ECC model of the SiC/Si iHPM. The ECC model can be split into four components: the output stage, the three-phase inverter, the three-phase rectifier, and the AC voltage source. The same output stage and the three-phase inverter are used in both the inverter circuit, Figure 11b, and the Si/SiC iHPM, Figure 11c. The SVPWM with a switching frequency of 30 kHz is implemented for driving the three-phase inverter. Because of the increased switching frequency, there is a longer dead time of 2.5 μs. The AC voltage source provides the three-phase root mean square (RMS) voltage (V_L(rms)_) of 233 V with an input frequency of 60 Hz and a phase shift of 120 degrees. The three-phase rectifier comprises six Si diodes, eliminating the need for control signals. Moreover, to avoid excessive oscillations in the DC output voltage of the three-phase rectifier, a capacitor is added between the DC output of the three-phase rectifier and the DC input of the inverter to stabilize the DC bus voltage.

## 6. Result and Discussion

### 6.1. Performance Evaluation of the SiC Inverter in the iHPM System

#### 6.1.1. Parasitics Extraction via Electromagnetic Analysis

The extracted parasitic inductances of the current paths, P-O_1_, O_1_-N, P-O_2_, O_2_-N, P-O_3_, and O_3_-N (O_1_,O_2_,O_3_), of the inverter are presented in Table 2. It can be observed from this table that the upper arm of the W phase (P-O_3_) has the highest parasitic inductance value, whereas the lower arm of the U phase (O_1_-N) exhibits the lowest one. This may mainly be because the O_1_-N path has the shortest current path, while the P-O_3_ path possesses the longest one, as can be seen in Figure 10. Moreover, the parasitic inductances of the P-N paths of the three phases are all less than 20 nH, which suggests that the current Cu trace layout design of the IMS substrate meets the design specification. Additionally, the parasitic inductances of the P-N paths of the three phases are all lower than those of the other conducting paths, i.e., P-O_i_ and O_i_-N (*i* = 1,2,3), indicating that the Cu trace layout design turns out to be fairly successful in suppressing the parasitic inductances through the mutual coupling. For validation, the parasitic inductances of the conducting paths, P-O_i_ and O_i_-N (*i* = 1,2,3), of these three phases are also characterized from the DPT experiment. Since the iHPM is configured in a three-phase bridge configuration, which shares the same positive (P) and negative (N) terminals, and also because the DUT is a power MOSFET body diode, which has no turning-off function, the loop parasitic inductance (P-N) cannot be obtained from the DPT experiment. Figure 12a,b display the measured turn-off waveforms of the body diodes of the power switches on the high side (P-O_1_) and the low side (O_1_-N) of the U phase, respectively. As can be seen in Figure 12a, the measured voltage step V_STEP_ and the rate of the current change in the conducting path P-O_1_ during the ON time of the U-phase upper arm are 16.6 V and 1.6 A/ns, respectively. Table 3 presents the measured voltage steps, the rates of change in the current, and parasitic inductance values associated with these six current conducting paths in the three-phase SiC inverter. By comparing the simulated (Table 2) and measured parasitic inductance (Table 3), we detect that there is a very good consistency between them, with a maximum difference of only around 12.5% appearing at the P-O_1_. The maximum current carried by these Al wires ranges from 19 A to 22 A, which is less than their fusing current by around 29 A, according to the vendor data. Clearly, the currently developed iHPM remains within a safe range, primarily because the given load current condition is 200 A, which is the rated current value of the module.

#### 6.1.2. Characterization of DPT Switching Characteristics

The modeled transient waveforms of the SiC/Si iHPM from the DPT simulation using the ECC model are presented in Figure 13. In the figure, the blue solid line and the red solid line represent the simulated and experimental drain-source voltage waveforms, and the blue dotted line and the red dotted line represent the drain current waveforms, respectively. It is evident from the figure that, despite some differences on the oscillation frequency of the current and the voltage, the simulated and experimental transient waveforms exhibit considerable consistency. The oscillation frequency of the voltage and the current is principally influenced by the parasitic inductance and capacitance in the circuit. The fair deviations in the oscillation frequency of the current and the voltage waveforms between the simulation and the measurement can be mainly attributed to the external measurement-induced parasitics from the instrument, connection wires, and gate driver board. Moreover, it is worth mentioning that the peak values of the voltage and current transient waveforms remain within the rated values of the SiC MOSFETs, which suggests that there is no need for either an RC snubber to protect the devices nor an improved internal circuit design to reduce the parasitic parameters. The modeled and measured switching characteristics, namely the turn-on rise time (t_r_), fall time (t_f_), turn-on energy (E_on_), and turn-off energy (E_off_), are listed in Table 4. It is noticeable that the simulated results are in good agreement with the measured data.

An experimental performance comparison was also performed between the SiC MOSFET inverter system inside the iHPM and the Si IGBT inverter system inside a commercial iPM in terms of the system power loss and efficiency. Figure 14 displays the calculated transient waveforms of the SiC MOSFET and Si IGBT inverters during the turn-on and turn-off transients, and the corresponding measured turn-on and turn-off energy loss are listed in Table 5. Clearly, as seen in Figure 14, the SiC MOSFET inverter enjoys a faster switching transient than the Si IGBT inverter, thereby exhibiting a much lower switching loss, especially in the turn-on process.

#### 6.1.3. Comparison of Modulating Signal and Output Current with SVPWM/SPWM and Delta/Star Load Connections

SVPWM and SPWM are two of the most widely applied modulation schemes to control the output load voltages and currents for VFD applications. In this work, both are implemented to drive the voltage source three-phase inverter. Figure 15a summarizes the time functions of the generated SPWM and SVPWM modulating signals. As seen in the figure, the modulating signal for SVPWM has a wider wave crest compared with that of SPWM, thereby achieving more output voltage and better DC link voltage utilization than the SPWM control. The U-phase output line current of the inverter with SVPWM and SPWM controls are reported in Figure 15b. The red and blue lines represent the output line currents in a delta-connected load with SVPWM and SPWM control. The results illustrate that the RMS output line current (I_L(rms)_) are 30.4 A and 25.7 A for the SVPWM and SPWM controls, respectively. The output line current for the SVPWM control is slightly higher than that of the SPWM control, indicating that the SVPWM control yields higher output power under the same conditions. Moreover, Figure 15b also presents the output line current of the star-connected inverter with the SVPWM control. With the same SVPWM control signal, the peak output line current drawn by a delta-connected load is approximately three times that of a star-connected load.

#### 6.1.4. Open-Loop Inverter Operation and Power Loss Estimation

The open-loop three-phase output current and voltage and power loss of the SiC inverter in the SiC/Si iHPM are further examined using the proposed ECC model. The simulated output voltage and current waveforms of the three-phase inverter are given in Figure 16, which are displaced through a 120-degree phase shift with each other. The simulation employs the same DC voltage of 404 V and ambient condition (room temperature) as the experiment, giving rise to a DC current of around 28.8 A. With this DC voltage and current, the input power is estimated to be about 11.6 kW. The RMS line voltage (V_L(rms)_) and current (I_L(rms)_) are calculated as 210.6 V and 30.4 A, respectively, with which the output power is computed at around 11.1 kW. Based on the ratio of the output power and the input power, the efficiency of the inverter is determined to be around 95.5%.

For validation of the simulation results, the U-phase output line current and the voltage and power loss of the SiC inverter with the SVPWM in an open-loop operation are experimentally characterized, and the results are depicted in Figure 17. There is a very consistency in waveform frequency between the simulation and experiment, but a slight difference in the amplitude of the output waveforms. This minor deviation may be also attributed to measurement-induced parasitics. In other words, the SVPWM control enables the production of pure sinusoidal waveforms of output voltage and current. The produced DC current from the SiC inverter under the 404 V DC voltage and room temperature experimental condition is around 30.3 A, with which the input power is computed at about 12.2 kW. The oscilloscope indicates that the measured V_L(rms)_ and I_L(rms)_ are 221.15 V and 30.49 A, respectively, with which the AC output power is calculated as approximately 11.7 kW. The conversion efficiency of the SiC inverter in the iHPM is 95.7%, which agrees well with the simulation result, i.e., 95.5%.

Table 6 shows the estimated conduction, switching, and body diode losses of these six SiC power switches in the SiC inverter, i.e., S1, S2, S3, S4, S5, and S6, as reported in Figure 10a, as well as their sum during a three-phase switching operation under a 404 V DC voltage, a 28.8 A load current, and room temperature conditions. It is noted that each of these switches consists of two SiC MOSFET devices, and the data in this table denote the average power loss values over one output cycle. The table presents that these six switches have a very comparable summed power loss, with an average value of around 23.58 W. The minor difference mainly arises from the varying current path lengths inherent in the inverter design. The ratios of the switch conduction, switching losses, and the body diode loss to their sum are 25.2%, 29.1%, and 45.6%. Among these three losses, the body diode loss is the maximum, and the switching loss surpasses the conduction loss. This result is very consistent with the literature for an Si power MOSFET inverter [14]. Furthermore, the total power loss of the inverter (six power switches) reaches as much as 141.51 W.

The system power losses of the SiC MOSFET inverter system inside the iHPM and the Si IGBT inverter system inside the commercial iPM during an open-loop operation are experimentally characterized under similar load conditions to those used in the open-loop operation experiment for the three-phase SiC inverter in Section 3.3, with identical operating and environmental conditions. Note that some of load conditions, including the DC bust voltage (V_dc_), the DC bus current (I_dc_) input power, the RMS line voltage (V_L(rms)_) and current (I_L(rms)_), and the output power are displayed in Table 7. The total system losses of the SiC MOSFET and the Si IGBT inverters are also listed in Table 7, which are 1063 W and 561.1 W, respectively. The corresponding efficiencies are calculated as 91.55% and 95.42%. The SiC MOSFET inverter turns out to have a superior electrical performance to the Si IGBT inverter.

### 6.2. Power Loss Prediction of the iHPM System

The electrical behavior and power loss of the SiC/Si iHPM during operation are evaluated through the proposed ECC model, as displayed in Figure 11c, which combines the functionalities of the three-phase rectifier and SiC inverter. For the open-loop operation of the three-phase iHPM, the switching frequency starts with 15 kHz. The other load conditions, such as the output power, the modulation index, the duty cycle, the dead time, and the output frequency, are identical to those used in the open-loop operation of the three-phase SiC inverter. In order to fulfill the 12 kW output power demand, the DC input voltage of the inverter needs to be maintained at 404 V. By using the formula for converting AC to DC voltage, the input line voltage for each phase is estimated to approximately 233 V to maintain 404 V DC bus voltage. With this input line voltage (i.e., 233 V), the generated DC output waveform from the three-phase rectifier from the three-phase (R, S, and T) AC voltage source is reported in Figure 18. It is clearly seen that the average DC voltage output from the three-phase rectifier is only about 396 V with a fluctuation of around 5 V. This by no means meet the goal, i.e., a DC bus voltage of 404 V, and this mainly arises from the diode losses of the rectifier and the parasitic parameters, and suggests that the input line voltage needs to be appropriately tuned. As the input line voltage is increased from 233 V to 238 V, the average output DC voltage can reach 404 V with the same fluctuation level, as also demonstrated in Figure 18.

In Figure 18, a fair voltage fluctuation of about 5 V in the output DC bus voltage can be observed. It is essential to find out how this voltage fluctuation would affect the output voltage and the current of the inverter. The predicted inverter output voltage and current waveforms using the SiC inverter and the SiC/Si iHPM ECC models under the same three-phase operation conditions are displayed in Figure 19. It is worthwhile mentioning that, for the SiC inverter ECC model, a constant DC voltage of 404 V without the voltage fluctuation is applied. As can be seen in Figure 19a, there is an imperceptibly slight variation in the output voltage and the current waveforms between these two models, indicating that the voltage fluctuation has a trivial impact on the output voltage and current.

The predicted total power losses of the Si rectifier, SiC inverter, and iHPM are around 85.32 W, 144.4 W, and 239.49 W. The remaining 9.77 W power loss of the iHPM can be ascribed to the conduction, switching, and body diode losses arising from the oscillation of the DC input voltage. The specification of the switching frequency for the iHPM is 30 kHz, and the dead time is re-set to 2.5 μs. The electrical behavior and power loss of the SiC/Si iHPM during the three-phase operation under the aforementioned load conditions are also assessed. Figure 20 presents the waveforms of the U-phase output voltage and current at the switching frequencies of 15 kHz and 30 kHz. The amplitudes of the output voltage and the current waveforms considerably diminish from 615 V to 537 V and from 86 A to 73 A, respectively, and the corresponding output power is reduced from 11.4 kW to 8.4 kW. The reduction in the output power is largely due to an increased switching frequency, as shown in Figure 21, where the left vertical axis represents the average single switch losses (i.e., conduction, switching, and diode loss) and the single rectifier diode loss, and the right vertical axis denotes the total power loss of the iHPM, which is the sum of the inverter loss and rectifier loss. In addition, the other factors include the increase in the body diode loss as a result of a longer dead time. This design does not meet the output power specification, i.e., the output power cannot reach 12 kW. The output power can be elevated through adjustment of the duty cycle and the DC voltage, which, however, may in turn increase the power loss. This may be one of the most concerning issues in system design.

As the duty cycle increases from 82% to 98%, the output waveforms of the line voltage and the current of the three-phase SiC/Si iHPM are illustrated in Figure 22, where the values of the V_L(rms)_ and I_L(rms)_ are 219.9 V and 31.6 A, respectively. Noticeably, the output power is found to attain the target value of 12 kW at a duty cycle of 98%. Similarly, the corresponding total power loss also enlarges from 280.19 W to 305.99 W, as depicted in Figure 23, resulting from increased conduction, switching, and body diode losses. The individual inverter switch losses (conduction, switching, and body diode) as well as their sum, the individual rectifier diode loss, and the total power loss of the iHPM, are presented in Table 8. Based on Table 8, the average power losses of these inverter switches and rectifier diodes are 36.88 W and 14.12 W, respectively. Moreover, there is a very comparable power loss across the inverter switches and rectifier diodes. The average switching, conduction, and body diode losses of the inverter switches are approximately 45.6%, 21.0%, and 33.4% of their sum (i.e., the total power loss in a single switch), suggesting that the switching loss is at its maximum at the specified switching frequency, followed by the body diode loss and the conduction loss. Moreover, the body diode loss is very comparable to the switch switching loss at the chosen switching frequency. The cause of the significant body diode loss can be mainly attributable to the high switching frequency (30 kHz) and gate resistance (13 Ω). Of the total power loss of the iHPM, i.e., 305.99 W, the SiC inverter accounts for about 72.3% and the rectifier takes the remaining 27.7%.

## 7. Conclusions

In this study, for performance and cost considerations, a 30 kHz/12 kW SiC/Si iHPM that integrates a brake circuit, a three-phase Si rectifier, and a three-phase SiC inverter within a single package for VFD applications is presented with an inductance cancellation circuit to minimize the parasitic inductance effect. In addition, the dynamic characteristics, power loss, and efficiency of the iHPM controlled by SVPWM during operation are theoretically and experimentally evaluated using the proposed ECC model, and the simulation and experimental results are compared with each other. In addition, the developed SiC MOSFET inverter is compared with the Si IGBT inverter in terms of performance. A few important conclusions are drawn below:From the predicted parasitic inductance results of the P-N paths of three phases, it is found that the current Cu trace layout design of the IMS substrate can satisfy the design requirements and is also quite effective in diminishing the parasitic inductances through mutual coupling.The simulated parasitic inductances are found to have a good agreement with the measured results, validating the proposed EA model. In addition, the simulated switching characteristics match well with the experimental results, demonstrating the effectiveness of the proposed ECC model.The switching transient performance of the SiC MOSFET inverter is remarkably superior to that of the Si IGBT inverter, causing a considerably reduced switching loss, especially in the turn-on process.The SVPWM control gives a marginally larger output line current than the SPWM control, thereby resulting in a greater output power under the same conditions.The delta load connection is shown to create a peak output line current approximately three times greater than the star load connection.The SiC inverter is found to outperform the Si IGBT inverter in both power loss and efficiency.A certain voltage oscillation is detected in the output DC bus voltage, and the voltage fluctuation is proven to have a marginal influence on the output voltage and current.As the switching frequency increase from 15 kHz and 30 kHz, the output power is greatly reduced, primarily because of the increase in switching loss and body diode loss arising from an increased switching frequency and a longer dead time, respectively.The duty cycle is an effective means to upgrade the output power loss of the iHPM during an open-loop operation.For the three-phase iHPM operated at a switching frequency of 30 kHz, a DC voltage of 404 V, a load current of 28.8 A, and a room temperature condition, the switch switching loss is at its maximum at the chosen switching frequency, followed by the body diode loss and the switch conduction loss. The high switching frequency (30 kHz) and gate resistance (13 Ω) are the main causes of the extensive body diode loss. Moreover, the SiC inverter loss dominates the total power loss of the iHPM.

## Figures and Tables

**Figure 1 micromachines-14-01356-f001:**
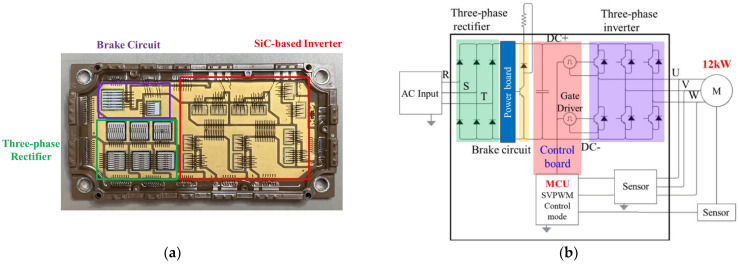
1200 V/200 A SiC/Si iHPM: (**a**) Prototype; (**b**) Electrical layout of the iHPM.

**Figure 2 micromachines-14-01356-f002:**
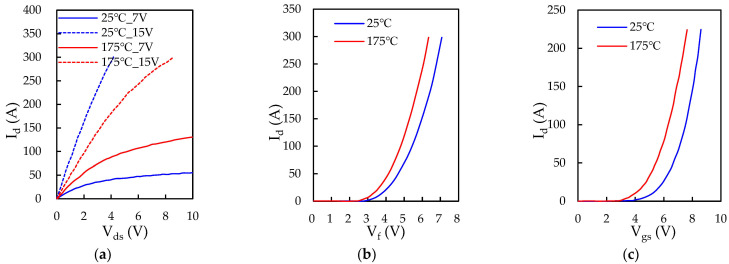
Characteristics of the SiC power MOSFETs: (**a**) Output characteristic; (**b**) Diode characteristic; (**c**) Transfer characteristic.

**Figure 3 micromachines-14-01356-f003:**
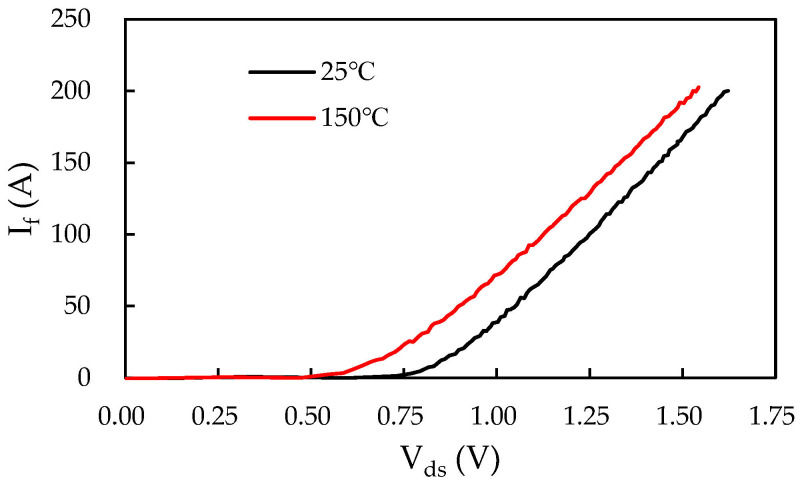
Characteristics of the rectifier Si diodes.

**Figure 4 micromachines-14-01356-f004:**
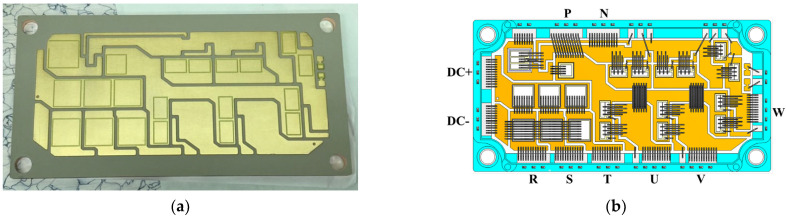
IMS and pin assignment and Cu layout design: (**a**) IMS for the SiC/Si iHPM; (**b**) Pin assignment and layout design.

**Figure 5 micromachines-14-01356-f005:**
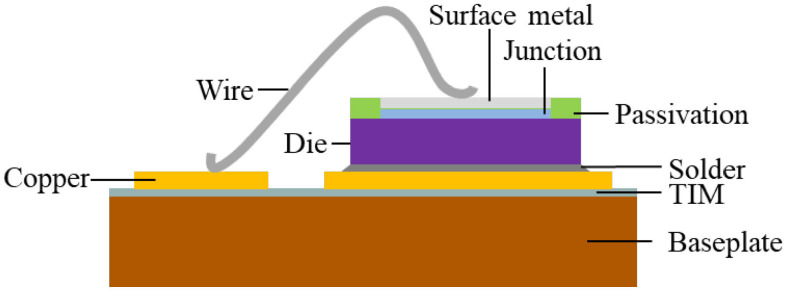
Schematic diagram of a cross-sectional view of a portion of the iHPM structure near the SiC inverter.

**Figure 6 micromachines-14-01356-f006:**
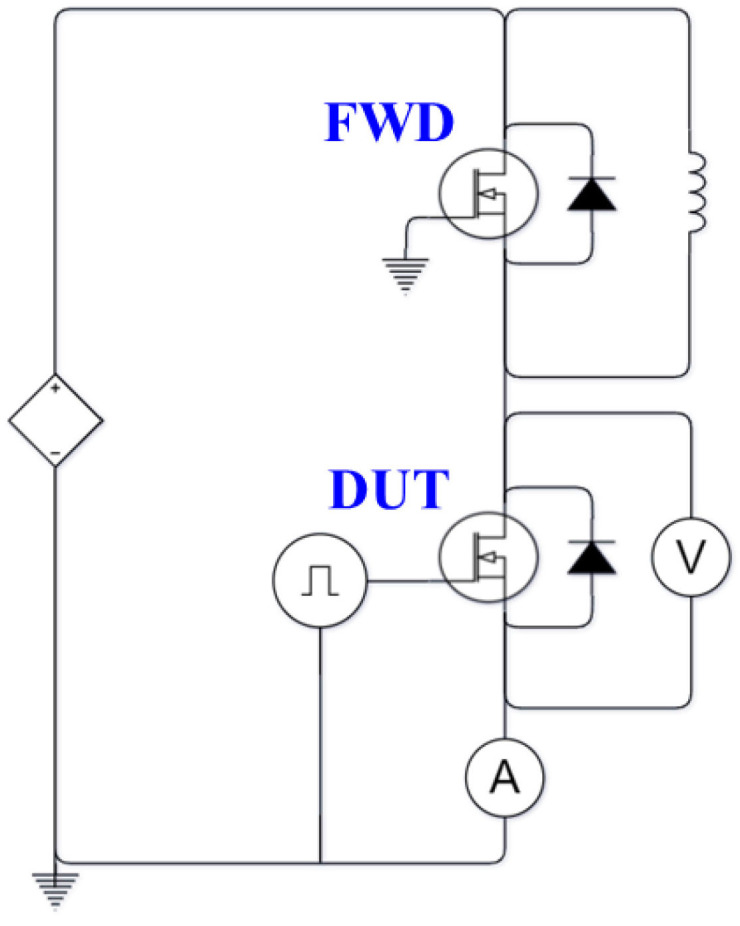
Measurement circuit of DPT.

**Figure 7 micromachines-14-01356-f007:**
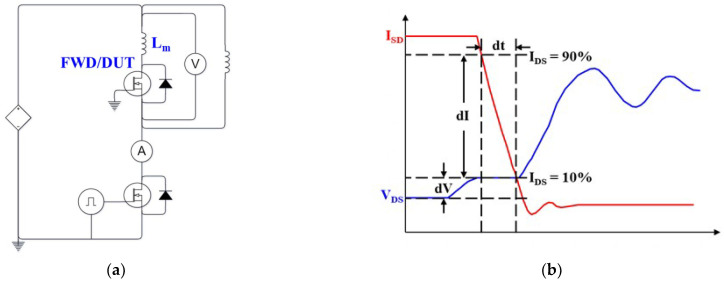
Inductance measurement: (**a**) Measurement circuit; (**b**) Schematic diagram of a typical switching waveform at FWD turn-off.

**Figure 8 micromachines-14-01356-f008:**
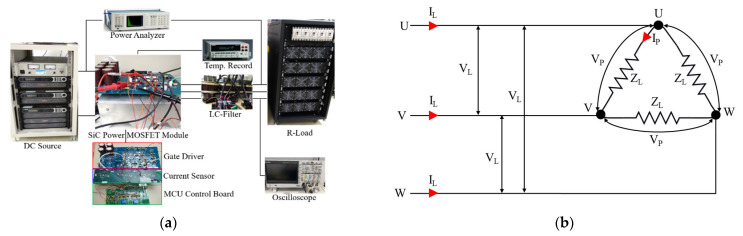
Open-loop inverter operation experiment: (**a**) Configuration; (**b**) Delta load connection.

**Figure 9 micromachines-14-01356-f009:**
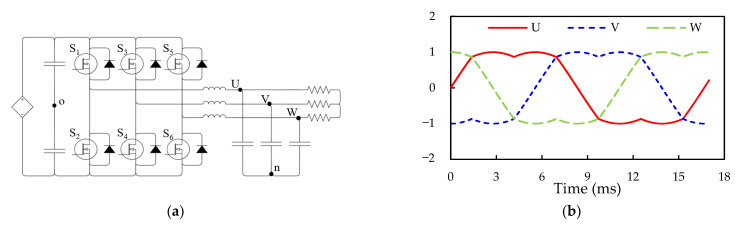
Space Vector Pulse Width Modulation: (**a**) Circuit; (**b**) Modulation waveforms of phase U, V, and W.

**Figure 10 micromachines-14-01356-f010:**
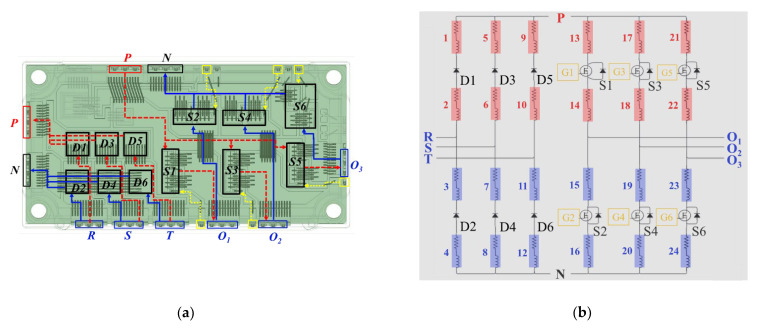
Current paths of the SiC/Si iHPM: (**a**) Geometry and current path of iHPM; (**b**) Circuit of iHPM with parasitics.

**Figure 11 micromachines-14-01356-f011:**
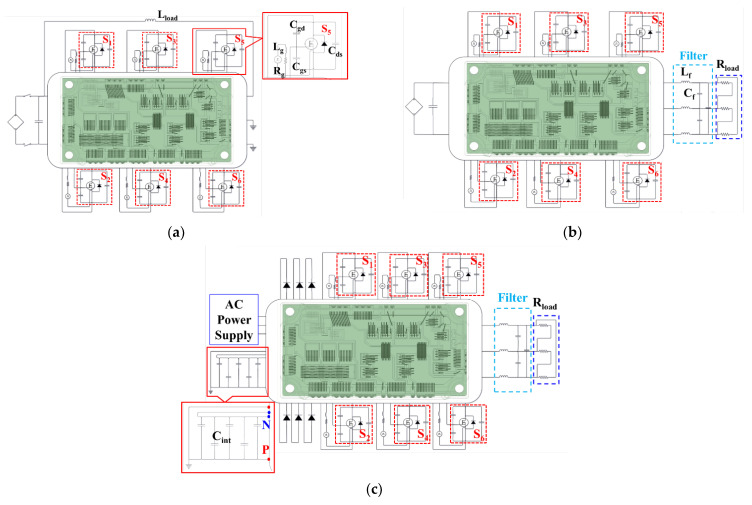
Proposed ECC model for electrical characterization and power loss estimate: (**a**) DPT; (**b**) SiC inverter; (**c**) SiC/Si iHPM.

**Figure 12 micromachines-14-01356-f012:**
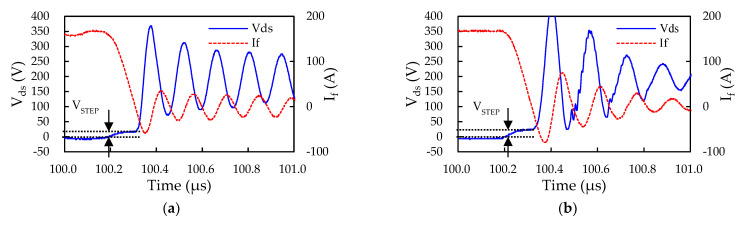
Measured current and voltage waveforms from parasitic inductance experiment: (**a**) High side of U phase; (**b**) Low side of U phase.

**Figure 13 micromachines-14-01356-f013:**
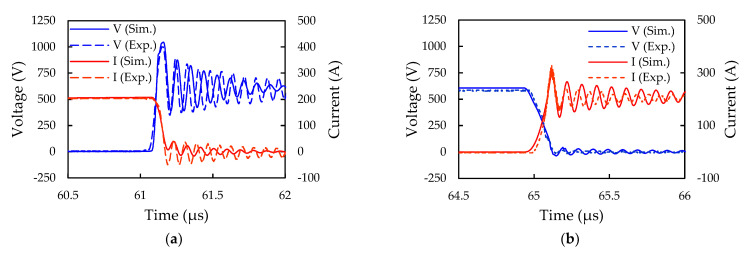
Modeled and measured voltage and current transient waveforms of the SiC/Si iHPM: (**a**) Turn-on; (**b**) Turn-off.

**Figure 14 micromachines-14-01356-f014:**
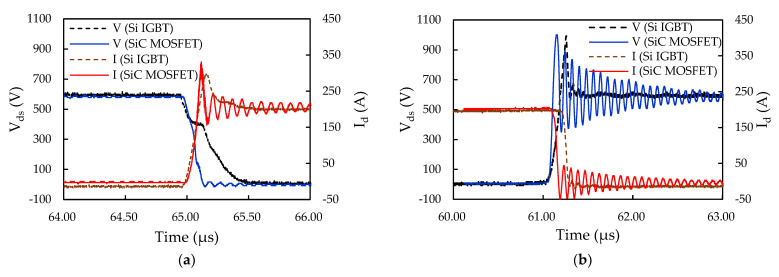
Measured transient waveforms of the SiC MOSFET and Si IGBT inverters: (**a**) Turn-on; (**b**) Turn-off.

**Figure 15 micromachines-14-01356-f015:**
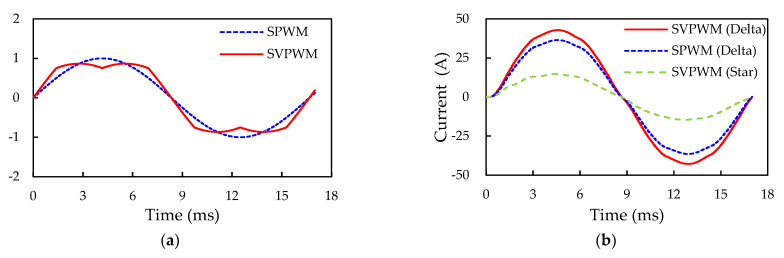
U-phase modulating signals for SPWM and SVPWM and the corresponding output line current waveforms under two different loading types: (**a**) Modulating signals; (**b**) Output line current.

**Figure 16 micromachines-14-01356-f016:**
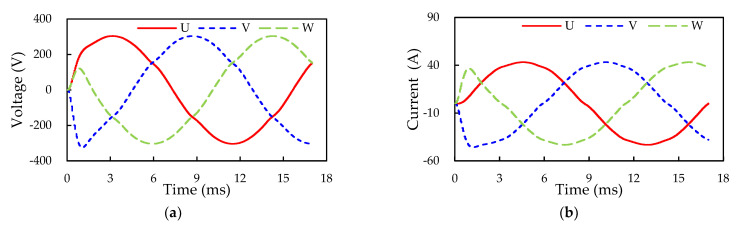
Three-phase inverter output voltage and current waveforms: (**a**) Output voltage; (**b**) Output current.

**Figure 17 micromachines-14-01356-f017:**
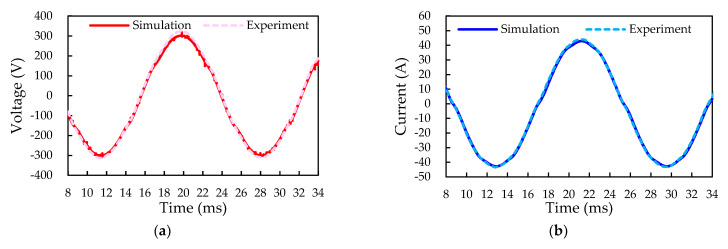
Measured and modeled U-phase output voltage and current waveforms: (**a**) Output voltage; (**b**) Output current.

**Figure 18 micromachines-14-01356-f018:**
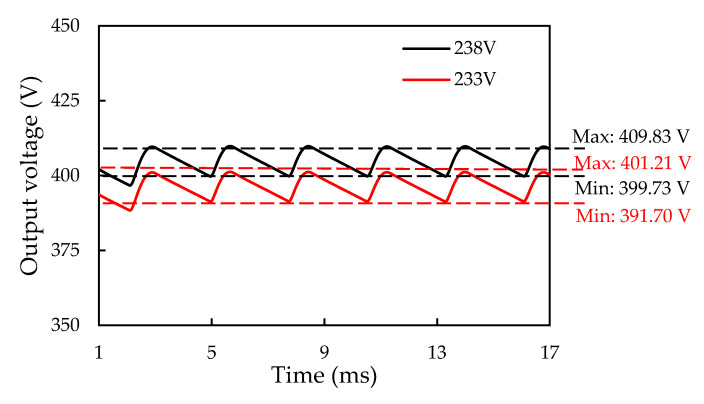
Output DC bus voltage of the three-phase Si rectifier for the 233 and 238 V input AC voltages: the dashed lines denote the maximum and minimum values of output DC bus voltages.

**Figure 19 micromachines-14-01356-f019:**
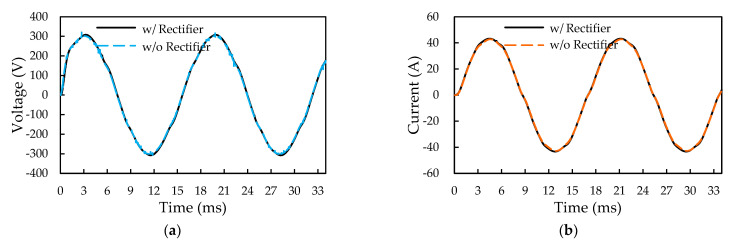
Effect of DC voltage fluctuation on the output voltage and current waveforms of the inverter: (**a**) Output voltage; (**b**) Output current.

**Figure 20 micromachines-14-01356-f020:**
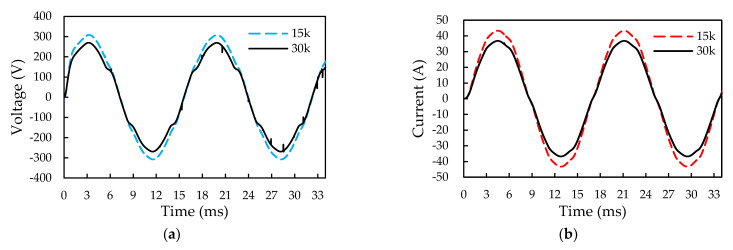
Output voltage and current waveforms at switching frequencies of 15 and 30 kHz: (**a**) Output voltage; (**b**) Output current.

**Figure 21 micromachines-14-01356-f021:**
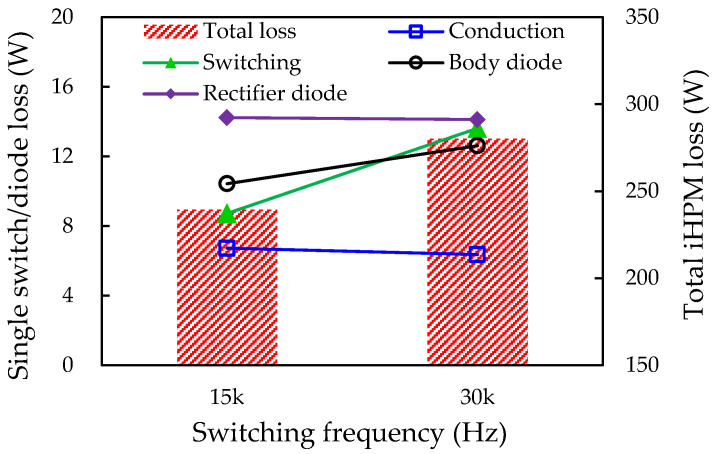
Effect of switching frequency on power loss of single switch and inverter.

**Figure 22 micromachines-14-01356-f022:**
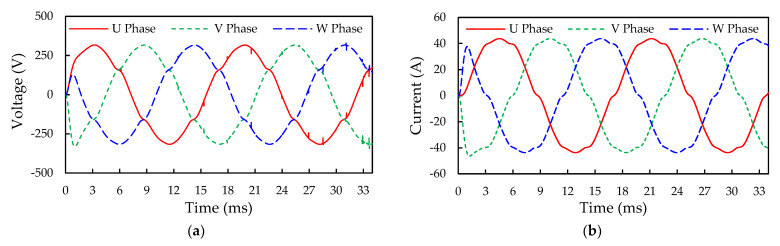
Three-phase output waveforms of the SiC/Si iHPM at a duty cycle of 98%: (**a**) Output voltage; (**b**) Output current.

**Figure 23 micromachines-14-01356-f023:**
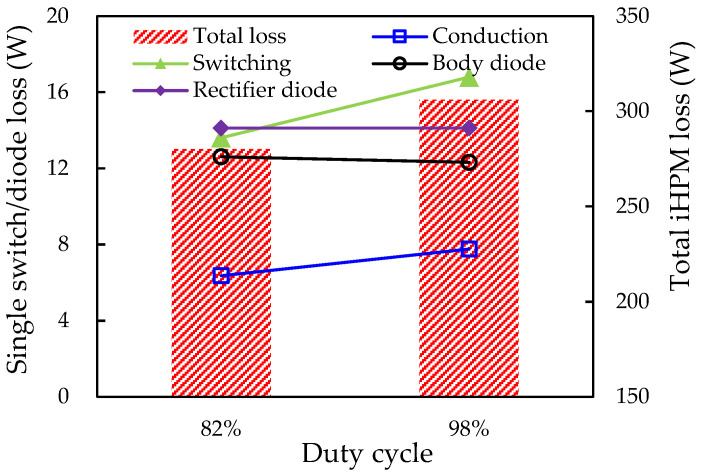
Effect of duty cycle on power loss.

**Table 1 micromachines-14-01356-t001:** Dimension of the integrated SiC power module.

	Size_X (mm)	Size_Y (mm)	Thickness (mm)
MOSFET	7.26	4.36	0.18
Rectifier	9	9	0.3
IGBT	8.8	8.8	0.14
Diode	6.26	6.26	0.35
Top Cu	-	-	0.3
TIM	119.1	59.1	0.12
Baseplate	119.1	59.1	3
Housing	122	62	-

**Table 2 micromachines-14-01356-t002:** Simulation results of the parasitic inductance.

Path	P-O_1_ (U)	O_1_-N (U)	P-O_2_ (V)	O_2_-N (V)	P-O_3_ (W)	O_3_-N (W)
L (nH)	34.62	31.49	41.15	36.63	47.00	39.42

**Table 3 micromachines-14-01356-t003:** Results of inductance measurement.

Path	P-O_1_ (U)	O_1_-N (U)	P-O_2_ (V)	O_2_-N (V)	P-O_3_ (W)	O_3_-N (W)
dv (V)	16.60	25.00	20.80	25.00	28.10	22.90
di/dt (A/ns)	1.67	1.79	1.61	1.65	1.51	1.63
L_m_ (nH)	39.60	34.63	44.22	41.29	50.23	41.66

**Table 4 micromachines-14-01356-t004:** Comparison of DPT switching characteristics between simulation and experiment.

	t_r_ (ns)	E_on_ (μJ)	t_f_ (ns)	E_off_ (μJ)
Simulation	140	2960	30	8003
Experiment	138	2836	25	8266
Difference	1.4%	4.2%	16.7%	3.3%

**Table 5 micromachines-14-01356-t005:** Switching energy losses of the SiC MOSFET and Si IGBT inverters.

	E_on_ (μJ)	E_off_ (μJ)
SiC MOSFET inverter	2836	8266
Si IGBT inverter	15,872	14,473

**Table 6 micromachines-14-01356-t006:** Inverter switch power losses and their sum (unit: W).

	Conduction	Switching	Body Diode	Sum	Total Loss of SiC Inverter
S1	5.33	7.70	10.47	23.50	141.51
S2	5.45	6.20	11.57	23.22	
S3	6.20	7.34	10.37	23.91	
S4	6.08	6.70	10.62	23.40	
S5	6.82	6.73	10.66	24.21	
S6	5.83	6.55	10.87	23.25	
Average	5.95	6.87	10.76	23.58	

**Table 7 micromachines-14-01356-t007:** Measured efficiencies of the SiC MOSFET and Si IGBT inverters.

	Si IGBT PM	SiC MOSFET Inverter
V_dc_ (V)	404.67	404.07
I_dc_ (A)	31.09	30.3
Input power	12,580.79	12,241.3
V_L(rms)_ (V)	220.71	221.15
I_L(rms)_ (A)	30.13	30.49
Output power (W)	11,517.79	11,680.17
Total system loss (W)	1063	561.1
System efficiency	91.55%	95.42%

**Table 8 micromachines-14-01356-t008:** Inverter switch losses and rectifier diode losses in the SiC/Si iHPM during open-loop operation (unit: W).

Switch/ Diode	Inverter Switch Loss				Rectifier Diode Loss	Total Loss
	Conduction	Switching	Body Diode	Sum		
S1/D1	7.30	18.43	11.23	36.96	14.12	305.99
S2/D2	6.94	16.29	12.41	35.64	14.12	
S3/D3	8.05	16.40	11.48	35.93	14.12	
S4/D4	7.74	15.99	14.51	38.24	14.12	
S5/D5	8.98	18.26	13.21	40.45	14.12	
S6/D6	7.56	15.42	11.08	34.06	14.12	
Avg.	7.76	16.80	12.32	36.88	14.12	

## Data Availability

Data sharing is not applicable.

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
