# Peer review of "Development and Performance Evaluation of Integrated Hybrid Power Module for Three-Phase Servo Motor Applications"

_micromachines, 2023, doi:10.3390/mi14071356_

Round 1

Reviewer 1 Report

This study developed a silicon carbide (SiC)/Si integrated hybrid power module (iHPM)it can be easily used to industrial servo motor, and it has practical value and important significance. Various experimental studies have been conducted on the hybrid power module, it shows that the design satisfies the design requirements.

Reviewer 2 Report

Autors comments.

1. how have you selected the output freqnency of 30 kHz? Is it an optimum freqerency.

2. In Fig. 18, what is factors or circuit elements that determine the ripple voltage in more details.

3. In tables 6 and 8. explain how to estimate or simulate each loss component. 

The article is minor editing of English language requied.